# Characterization and Evaluation of Layered Bi_2_WO_6_ Nanosheets as a New Antibacterial Agent

**DOI:** 10.3390/antibiotics10091068

**Published:** 2021-09-03

**Authors:** Maria Karollyna do Nascimento Silva Leandro, João Victor Barbosa Moura, Paulo de Tarso Cavalcante Freire, Maria Leticia Vega, Cleânio da Luz Lima, Ángel Alberto Hidalgo, Ana Carolina Justino de Araújo, Priscilla Ramos Freitas, Cicera Laura Roque Paulo, Amanda Karine de Sousa, Janaina Esmeraldo Rocha, Lívia Maria Garcia Leandro, Rakel Olinda Macedo da Silva, Natália Cruz-Martins, Henrique Douglas Melo Coutinho

**Affiliations:** 1Department of Biological Chemistry, Regional University of Cariri, Crato 63105-000, Brazil; karollynasilva@leaosampaio.edu.br (M.K.d.N.S.L.); caroljustino@outlook.com (A.C.J.d.A.); priscilla.r.freitas@hotmail.com (P.R.F.); lauraroquealencar@gmail.com (C.L.R.P.); amandakarine@leaosampaio.edu.br (A.K.d.S.); janainaesmeraldo@gmail.com (J.E.R.); 2Department of Biomedicine, University Center Dr. Leão Sampaio, Juazeiro do Norte 63040-005, Brazil; liviamariagarcialeandro@hotmail.com (L.M.G.L.); rakelolinda@leaosampaio.edu.br (R.O.M.d.S.); 3Department of Physics, Science and Technology Center, Federal University of Maranhão, São Luís 65085-580, Brazil; jvb.moura@ufma.br; 4Department of Physics, Campus do Pici, Federal University of Ceará, Fortaleza 60455-760, Brazil; tarso@fisica.ufc.br; 5Department of Physics, Campus Ministro Petrônio Portella, Federal University of Piauí, Teresina 64049-550, Brazil; marialeticia.vega@gmail.com (M.L.V.); cleanio@ufpi.edu.br (C.d.L.L.); angel.ufu@gmail.com (Á.A.H.); 6Faculty of Medicine, University of Porto, Alameda Prof. Hernani Monteiro, 4200-319 Porto, Portugal; 7Institute for research and Innovation in Health (i3S), University of Porto, Rua Alfredo Allen, 4200-319 Porto, Portugal; 8Institute of Research and Advanced Training in Health Sciences and Technologies (CESPU), Rua Central de Gandra, 1317, 4585-116 Gandra, Portugal

**Keywords:** bismuth tungstate, structural properties, antibacterial agent, antibiotic resistance

## Abstract

**Background**: Pathogenic microorganisms are causing increasing cases of mortality and morbidity, along with alarming rates of ineffectiveness as a result of acquired antimicrobial resistance. Bi_2_WO_6_ showed good potential to be used as an antibacterial substance when exposed to visible light. This study demonstrates for the first time the dimension-dependent antibacterial activity of layered Bi_2_WO_6_ nanosheets. **Materials and methods**: The synthesized layered Bi_2_WO_6_ nanosheets were prepared by the hydrothermal method and characterized by powder X-ray diffraction (XRD), scanning electron microscopy (SEM), atomic force microscopy (AFM), and Raman and Fourier transform infrared spectroscopy (FTIR). Antibacterial and antibiotic-modulation activities were performed in triplicate by the microdilution method associated with visible light irradiation (LEDs). **Results**: Bi_2_WO_6_ nanosheets were effective against all types of bacteria tested, with MIC values of 256 μg/mL against *Escherichia coli* standard and resistant strains, and 256 μg/mL and 32 μg/mL against *Staphylococcus aureus* standard and resistant strains, respectively. Two-dimensional (2D) Bi_2_WO_6_ nanosheets showed antibacterial efficiency against both strains studied without the presence of light. **Conclusions**: Layered Bi_2_WO_6_ nanosheets revealed dimension-dependent antibacterial activity of the Bi_2_WO_6_ system.

## 1. Introduction

Microbial infections triggered by resistant pathogens have raised important public health problems in the world, being currently the focus of study by multiple researchers in an intent to provide safer, effective and less harmful antimicrobials [1]. Among the various fields under intense research, nanobiotechnology has been on high demand, with an increasing number of nanomaterials being highlighted in several scientific studies as a possible alternative to fight resistant microorganisms. Characterized by a very small size, but with a large surface area, such nanoformulations have revealed the capability of improving the delivery of drugs in specific tissues, ultimately boosting drugs’ effectiveness and a reduction in adverse effects, as both dose and time of action can be thoroughly controlled [2].

After the discovery of graphene, nanomaterials with a 2D dimension are increasingly being targeted by research seeking to elucidate physical properties in relation to their bulk precursors [3]. However, in the post-graphene era, numerous inorganic materials in the form of layers have been extensively investigated, such as transition metal dichalcogenides [4] and layered metal oxides [5], among others. Monolayered bismuth tungstate (Bi_2_WO_6_) nanosheets is a representative example; the Aurivillius oxide Bi_2_WO_6_ nanosheets has a sandwich substructure of [BiO]^+^—[WO_4_]^2−^—[BiO]^+^. Layered Bi_2_WO_6_ nanosheets were obtained by a cetyltrimethylammonium bromide-assisted bottom-up route [6].

Bismuth tungstate (Bi_2_WO_6_) has received huge attention as a visible light photocatalyst [7]. Bi_2_WO_6_ has shown interesting photochemical stability and reusability, being potentially useful for environmental treatment purposes, while also exhibiting photocatalytic degradation of Erichrome Black T (EBT) organic dye [8]. Furthermore, Bi_2_WO_6_ exhibited photocatalytic activity for degrading norfloxacin and enrofloxacin under visible light irradiation, meaning its feasible application along with fluoroquinolone antibiotics [9]. In a study, Aurivillius oxide Bi_2_WO_6_ was applied as an effective visible light-driven antibacterial photocatalyst for *Escherichia coli’s* inactivation, where neither visible light without the photocatalyst nor Bi_2_WO_6_ in the dark revealed bactericidal effects. Thus, the bactericidal effect in *E. coli* was certainly attributed to the photocatalytic reaction of Bi_2_WO_6_ under visible light irradiation [10].

In this sense, the present study aims to assess, for the first time, the antibacterial and antibiotic modulation effects of layered Bi_2_WO_6_ nanosheets against the standard and multidrug-resistant (MDR) *Staphylococcus aureus* and *E. coli*. Two-dimensional (2D) Bi_2_WO_6_ nanosheets showed antibacterial efficiency in both studied strains (Gram-negative and Gram-positive bacteria) without the presence of light (in the dark), showing a dimension-dependent antibacterial activity of the Bi_2_WO_6_ system and improving its properties in relation to the bulk material.

## 2. Results and Discussion

### 2.1. Characterization of Layered Bi_2_WO_6_ Nanosheets

Figure 1 shows the XRD powder pattern of Bi_2_WO_6_ obtained by a CTAB-assisted hydrothermal method. The crystalline nature of the samples was confirmed, and all diffraction peaks can be indexed to an orthorhombic Bi_2_WO_6_ phase with lattice parameters a = 5.457 Å, b = 5.436 Å and c = 16.427 Å (JCPDS Card N° 73-2020), without secondary phase. The intensity ratio of the (020)/(200) Bragg peak to the (113) peak was higher than the standard value. In addition, the full width at half maximum (FWHM) of the (200)/(020) Bragg peak was narrower than that of the (113) peak. This analysis indicates a higher grain size of the synthesized bismuth tungstate along the (100) and (010) directions compared to the (001) direction [11]. The AFM images (Figure 2a,b) and the corresponding height histograms (Figure 2c,d) of the layered Bi_2_WO_6_ nanosheets show that the monolayer has a thickness of about 0.9 nm, which is in agreement with that of Bi_2_WO_6_ monolayer along the (001) direction. In addition, the three-dimensional (3D) morphology in Figure 2b indicates that the layered Bi_2_WO_6_ nanosheets have a dense and flat morphology. Therefore, the AFM result agrees with the XRD results. Furthermore, this result is in perfect agreement with data obtained by Zhou et al. [6], who showed by high resolution transmission electron microscopy (HRTEM) and atomic force microscopy (AFM) that layered Bi_2_WO_6_ nanosheets obtained by CTAB-assisted hydrothermal synthesis exposes (001) facets. In addition, the authors showed that layers had a thickness of 0.8 nm, i.e., the Bi_2_WO_6_ monolayer along the (001) direction features 1/2 the size of the unit cell in the c-axis direction.

As stated above and looking at Figure 3, Aurivillius Bi_2_WO_6_ is a layered material built up of [Bi_2_O_2_] layers and corner-shared WO_6_ octahedral layers, whereas the Bi_2_WO_6_ monolayer has a sandwich substructure configuration of [BiO]^+^—[WO_4_]^2−^—[BiO]^+^ exposing the Bi atoms on the surface. The morphology of the two-dimensional Bi_2_WO_6_ nanosheets can be shown by the SEM images (Figure 4), which clearly reveal Bi_2_WO_6_ exhibiting thin two-dimensional structures.

The room temperature Raman and Fourier transform infrared (FTIR) spectra of the synthesized layered Bi_2_WO_6_ nanosheets are presented in Figure 5 and Table 1. The Raman spectrum of Bi_2_WO_6_ monolayers in the spectra range between 100 and 900 cm^−1^ presents broad Raman peaks, some even overlapping forming only shoulders, making them difficult to distinguish in the unpolarized spectrum (See Figure 5a). On one hand, the Raman spectrum exhibits significant changes as the crystallite size decreases, comparing the Raman spectrum of Bi_2_WO_6_ monolayers with the spectrum of Bi_2_WO_6_ bulk structures. On the other hand, the Raman spectrum obtained is in full agreement with what is shown in the literature for nano-sized materials [12]. Looking at the FTIR spectrum in Figure 5b, we see that the broad band at 3460 cm^−1^ corresponds to the stretching vibration of O–H bonds, indicating water molecules adsorbed on the surface of the sample. The 2921 cm^−1^ and 2854 cm^−1^ are associated with the C–H stretching vibrations of the methyl and methylene groups of CTAB used in hydrothermal synthesis [13]. The assignment of the Raman and FTIR modes that come from the W–O or Bi–O bonds are listed in Table 1. In summary, the FTIR and Raman spectra obtained are in full agreement with what is shown in the literature and in agreement with the XRD results [12,14,15].

### 2.2. Antibacterial Activity of Layered Bi_2_WO_6_ Nanosheets

Bi_2_WO_6_ nanosheets revealed to be effective against all kinds of bacteria tested, with a MIC of 256 μg/mL against *E. coli* standard and resistant strains, and of 256 μg/mL and 32 μg/mL against *S. aureus* standard and resistant strains, respectively (Table 2). Evaluating the antibacterial activity of Bi_2_WO_6_ nanosheets under LED irradiation with varying wavelengths (415, 620 and 590 nm), no photocatalytic reaction was observed, given that MIC values do not decrease under visible light irradiation. Thus, Bi_2_WO_6_ nanosheets exhibit excellent antibacterial inactivation on *E. coli* and *S. aureus* in the dark.

In the study by Li et al. [16], the Bi_2_WO_6_ crystals in hierarchical flower-like morphology did not reveal antibacterial activity against *E. coli* in the dark, as some effect was only observed when the material was irradiated with visible light. Likewise, Hen et al. [10] using Aurivillius oxide Bi_2_WO_6_ as an effective visible light-driven antibacterial photocatalyst for *E. coli* inactivation, did not report any bactericidal effect under visible light without the photocatalyst nor Bi_2_WO_6_ in the dark. It is, however, worth noting that the results obtained by Hen et al. [10] and Li et al. [16] attributed the bactericidal effect to the photocatalytic reaction of Bi_2_WO_6_ (bulk) under visible light irradiation. In the present study, it was demonstrated that the antibacterial activity of the layered Bi_2_WO_6_ nanosheets was intrinsic to the material (no need for the presence of visible light).

Regarding the antibacterial effect of Bi_2_WO_6_ nanosheets, an intrinsic clinical relevance was found with the standard stated by Houghton et al. [17]. Another interesting point to underline is that this material affects Gram-positive and Gram-negative bacteria equally, which highlights the great clinical and technological potential of this material. This is especially important in view of the significant increase in resistance mechanisms developed by *E.coli* and *S. aureus*, such as the alteration of cell membrane permeability, making it difficult for antibiotics to enter and change the target drug binding sites, respectively [18,19].

In addition, the crystalline structure of layered Bi_2_WO_6_ nanosheets has the ability to generate e^−^–h^+^ pairs, while the sandwich substructure of [BiO]^+^—[WO_4_]^2−^—[BiO]^+^ monolayer Bi_2_WO_6_ simulates the heterojunction interface with space charge that promotes the separation of carriers generated in the interface [6]. As a result, such species interact with H_2_O, triggering the formation of OH^•^, H^+^, and O_2_^•−^, which can act on the cells’ surface to degrade some components of bacterial cell membrane. Cell rupture can cause the bacteria to lose function and eventually lead to cell death [20]. Moreover, the high surface area/volume ratio of 2D Bi_2_WO_6_ nanosheets favors the interaction between the sample and the bacterial membrane, facilitating possible adsorption processes. Therefore, based on the results discussed above, it is possible to observe the possibility that the mechanism responsible for the antibacterial activity of Bi_2_WO_6_ monolayers is related to the two-dimensional property of the system.

### 2.3. Modulation of Antibiotic Activity by Layered Bi_2_WO_6_ Nanosheets

Data obtained on the modulation of antibiotic activity demonstrated by Bi_2_WO_6_ nanosheets are shown in Figure 6. The compound significantly raised the MIC of both antibiotics (amikacin and gentamicin) against Gram-negative and Gram-positive strains, show no improvements in antibiotic activity, and the associations even resulted in antagonistic effects.

The association of aminoglycosides under visible light irradiation may represent a promising strategy in the treatment of skin infections caused by resistant bacteria. Therefore, in this study the antibiotic-modulating effects of LED light exposure associated or not with Bi_2_WO_6_ nanosheets was investigated (Figure 7). No potentiation of the action of aminoglycosides associated with Bi_2_WO_6_ nanosheets against Gram-positive and Gram-negative strains was stated, regardless of LED wavelength. Thus, the strong antagonistic effects observed when aminoglycosides were used in combination with Bi_2_WO_6_ nanosheets under visible light irradiation against Gram-positive and Gram-negative strains seem to be related to the high photocatalytic potential of bismuth tungstate in degrading organic compounds [8,9].

The emergence of antibiotic resistance by pathogenic bacteria occurs due to several factors and, among these, the inappropriate drugs’ disposal stands out. Specifically, wastewater from health facilities, mainly hospitals, stands out as an important source of emission of antibiotics in the environment and, thus, it is important to degrade these compounds before disposal. In this sense, current research is seeking new agents capable of acting as photocatalysts [21], and Bi_2_WO_6_ nanosheets showed a good potential in the treatment of effluents containing gentamicin and amikacin.

## 3. Materials and Methods

### 3.1. Synthesis

The synthesis of layered Bi_2_WO_6_ nanosheets was performed by a hydrothermal method, as described previously [6]. Sodium tungstate dihydrate [Na_2_WO_4_·2H_2_O] (≥99%, Sigma-Aldrich, St. Louis, MO, USA), bismuth nitrate pentahydrate [Bi(NO_3_)_3_·5H_2_O] (≥98.0%, Sigma-Aldrich, St. Louis, MO, USA) and hexadecyltrimethylammonium—CTAB [CH_3_(CH_2_)_15_N(Br)(CH_3_)_3_] were used as starting precursors. In a synthesis procedure, 1 mmol of Na_2_WO_4_·2H_2_O, 2 mmol of Bi(NO_3_)_3_·5H_2_O and 0.05 g of CTAB were dissolved in 80 mL of deionized water. This aqueous solution was stirred for 30 min at an average speed of 1500 rpm. The resulting solution was transferred to a 100 mL Teflon-lined stainless autoclave and maintained at 120 °C for 24 h. The white precipitates were repeatedly washed with deionized water and dried in an air oven at 60 °C for 10 h.

### 3.2. Structural Characterization

Structural characterization was performed by X-ray diffraction (XRD) using a Mini-Flex Rigaku diffractometer. Morphological analysis of layered Bi_2_WO_6_ nanosheets was carried out in a scanning electron microscope (SEM) model Vega3 Tescan. Atomic force microscopy images were recorded using NTMDT microscope. Fourier Transform Infrared (FT-IR) spectra were obtained using a Perkin Elmer Spectrum Two spectrophotometer. Raman measurements were performed by a Horiba LabRaman spectrometer.

### 3.3. Analysis of Antibacterial Activity and Antibiotic Resistance Modulation

The antibiotic-enhancing activity was assessed using the methodology of Coutinho et al. [22].

### 3.4. Experiments with LED Light Exposure

The Light Emitting Diodes-LED device (a light emitting diode; NEW Estética^®^, Fortaleza, Brazil) was used in the experimental protocols. The LEDs with a wavelength predetermined by the device used were blue (415 nm), red (620 nm) and yellow (590 nm). To assess the effect of LED light exposure on bacterial growth in modulating antibacterial activity, cultures and treatments were performed as described above. Plates were exposed to blue, red or yellow light for 20 min and then incubated at 37 °C for 24 h. Plates without exposure to LED light were used as experimental controls. Readings were performed as described above.

### 3.5. Statistical Analysis

Statistical analysis was performed using the GraphPad Prism 6.0 software, with an alpha set at 0.05. One-way analysis of variance (ANOVA) and Bonferroni’s post-hoc tests were used to address differences between groups. More details are shown in Appendix A.

## 4. Conclusions

Data obtained in this study provide significant insights into the dimension-dependent antibacterial activity of layered Bi_2_WO_6_ nanosheets. The crystalline nature of the samples was confirmed, and all diffraction peaks were indexed to the orthorhombic Bi_2_WO_6_ phase. Two-dimensional (2D) Bi_2_WO_6_ nanosheets showed antibacterial action against the strains studied without the presence of light, and also revealed a possible catalytic effect of antibiotics. Further studies are needed toward a more in-depth understanding on this action.

## Figures and Tables

**Figure 1 antibiotics-10-01068-f001:**
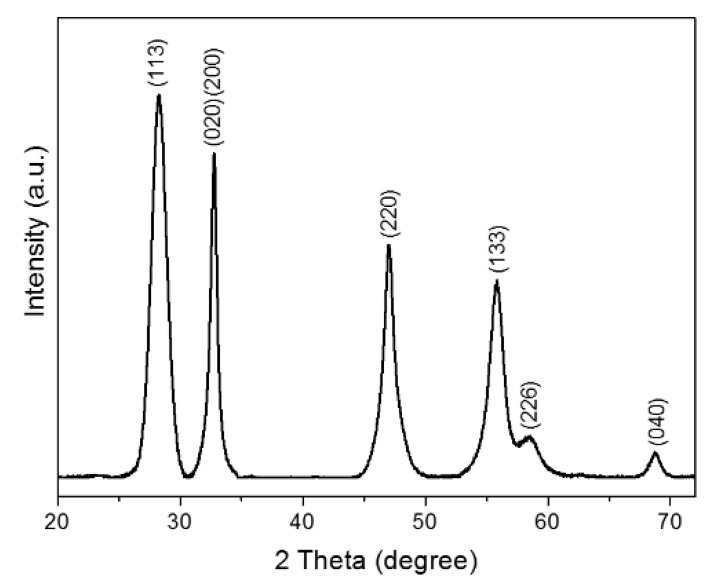
X-ray diffraction pattern of layered Bi_2_WO_6_ nanosheets obtained by CTAB-assisted hydrothermal synthesis.

**Figure 2 antibiotics-10-01068-f002:**
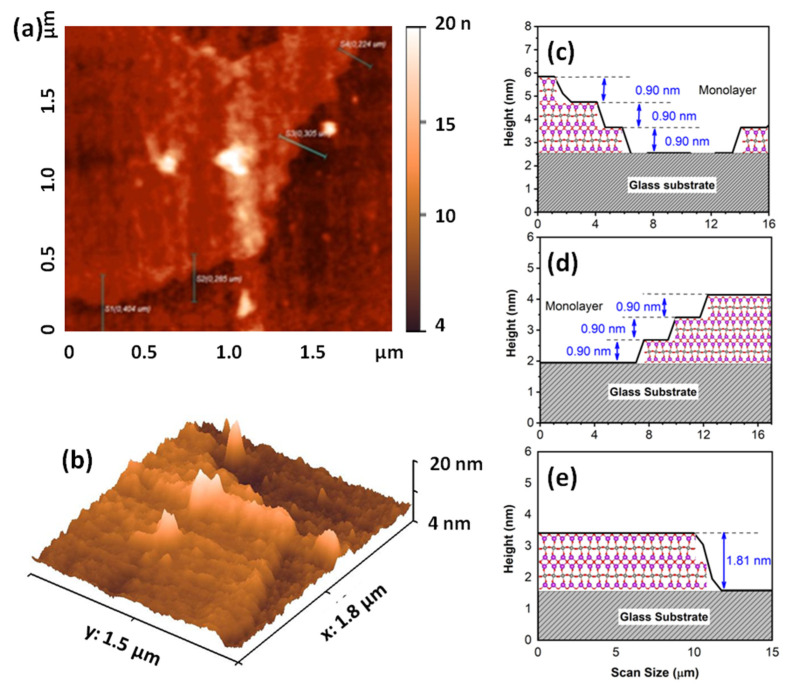
AFM images of the layered Bi_2_WO_6_ nanosheets: (**a**) Surface topography, (**b**) three-dimensional image and (**c**–**e**) the corresponding height histograms.

**Figure 3 antibiotics-10-01068-f003:**
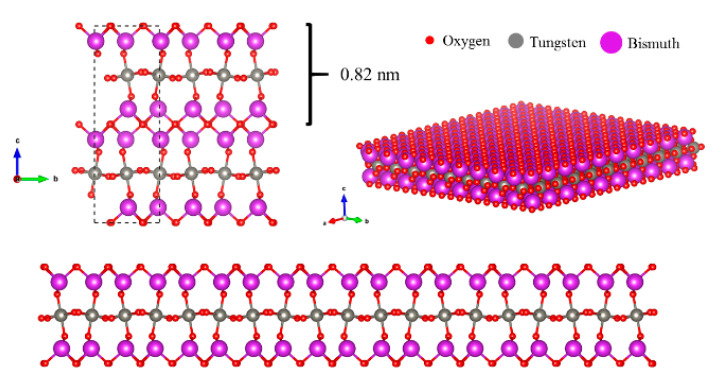
Bi_2_WO_6_ monolayer substructure configuration of [BiO]^+^—[WO_4_]^2−^—[BiO]^+^ exposing the Bi atoms on the surface.

**Figure 4 antibiotics-10-01068-f004:**
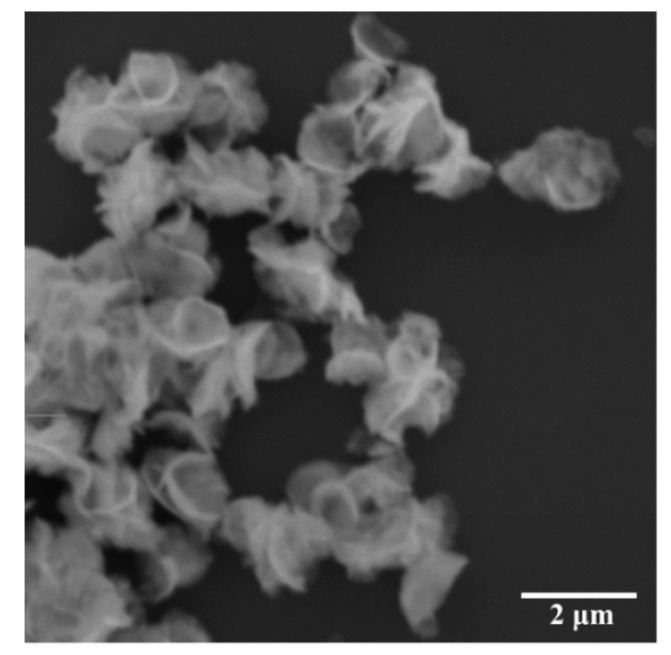
SEM image of layered Bi_2_WO_6_ nanosheets prepared by hydrothermal route showing two-dimensional structures.

**Figure 5 antibiotics-10-01068-f005:**
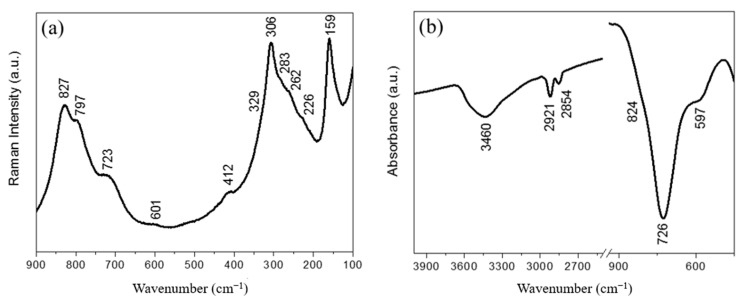
(**a**) Raman and (**b**) FTIR spectra of layered Bi_2_WO_6_ nanosheets at room temperature.

**Figure 6 antibiotics-10-01068-f006:**
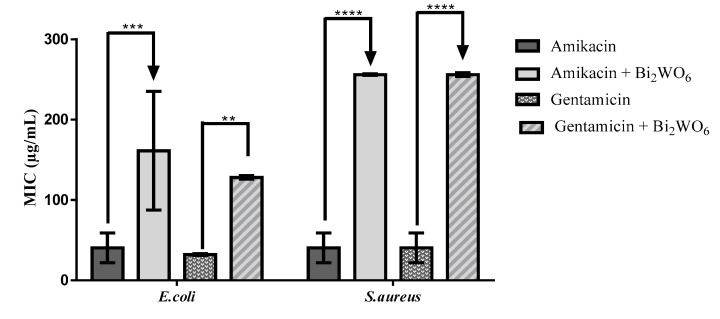
MIC of amikacin and gentamicin alone, or in the presence of Bi_2_WO_6_ monolayers, against multi-drug resistant strains of *S. aureus* and *E. coli*. **** *p* < 0.0001, *** *p* < 0.001, ** *p* < 0.01.

**Figure 7 antibiotics-10-01068-f007:**
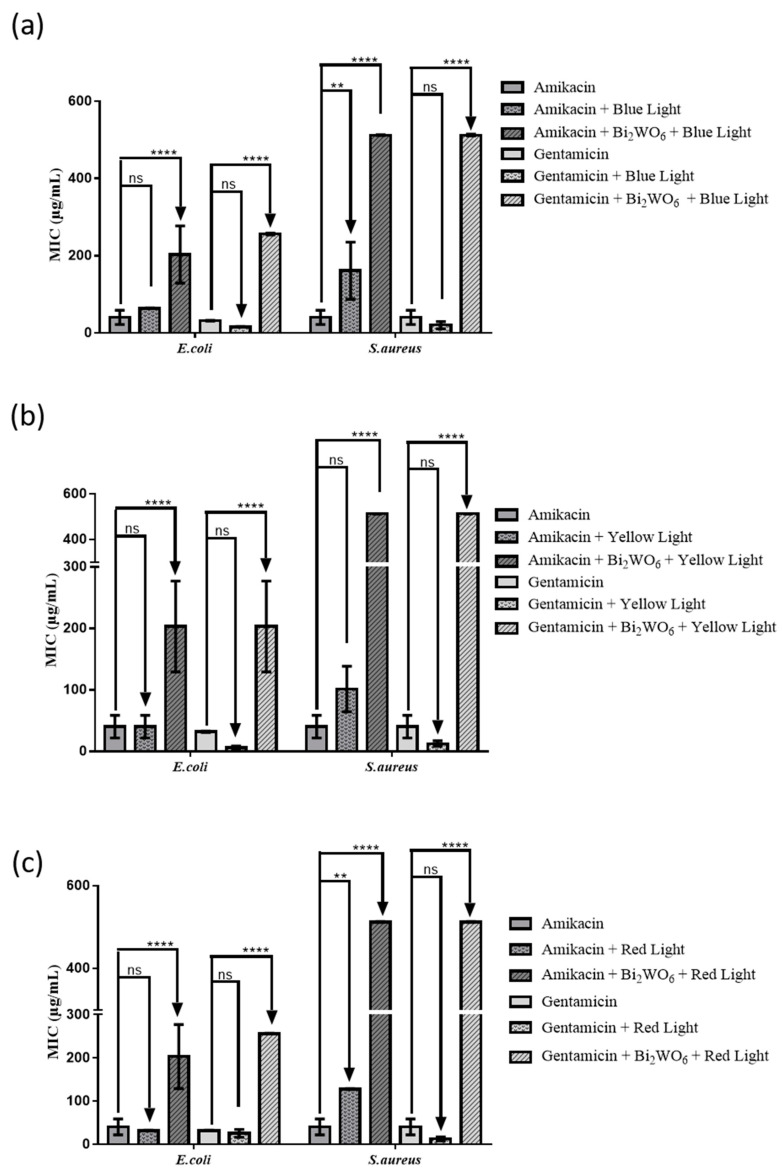
MIC of amikacin and gentamicin alone, or in the presence of Bi_2_WO_6_ monolayers and (**a**) blue, (**b**) yellow or (**c**) red LED lights, against MDR *S. aureus* and *E. coli* strains. **** *p* < 0.0001, ** *p* < 0.01, ns: value statistically non-significant.

**Table 1 antibiotics-10-01068-t001:** Observed Raman and FTIR modes of layered Bi_2_WO_6_ nanosheets.

Observed Modes (cm^−1^)		Assignment
Raman	Infrared	
827	824	Asymmetric stretching of WO_6_
797	-	Symmetric stretching of WO_6_
723	726	Asymmetric stretching of WO_6_
601	597	Bending of WO_6_
412	-	Bending of WO_6_
329	-	Bending of Bi-O bonds
306	-	Bending of WO_6_
283	-	Bending of WO_6_
262	-	Bending of WO_6_
226	-	Bending of WO_6_
159	-	Translational mode (Bi)

**Table 2 antibiotics-10-01068-t002:** MIC of layered Bi_2_WO_6_ nanosheets.

Treatment	E.C. ATCC 25922	S.A. ATCC 25923	E.C. 06	S.A. 10
Bi_2_WO_6_ monolayers	256	256	256	32
Bi_2_WO_6_ + Blue Light	341.3	256	256	32
Bi_2_WO_6_ + Yellow Light	256	256	512	64
Bi_2_WO_6_ + Red Light	256	256	256	32

S.A., *Staphylococcus aureus*; E.C., *Escherichia coli.* The MIC values are expressed in µg/mL.

## Data Availability

Data presented in this study are available on request from the corresponding author.

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
