# Peer review of "Characterization and Evaluation of Layered Bi2WO6 Nanosheets as a New Antibacterial Agent"

_antibiotics, 2021, doi:10.3390/antibiotics10091068_

Round 1

Reviewer 1 Report

First question is what sort of material have authors manufactured. Moreover,  manuscript uses not very precise language that is hard to understand.

As seen in AFM image (Fig. 2) the produced material is not “monolayer” but “multilayered” material oriented during deposition along the same crystallographic direction. On contrary to the original paper by Yangen Zhou from Nature Comm. 2015, authors have not provided any experimental evidence that it is in fact monolayer. Moreover, to reference to the someone else paper is not enough and authors should provide in the manuscript exact experimental procedure and experimental data to convince us what kind of material are they working with.

If this is not possible, since the deposited material has quite limited thickness and contains only several sandwich-type layers, probably the best description would be “nanolayered material”. Hydrothermal methods in general, allow to obtain homogenous monocrystalline multilayered solids of almost unified size and the structure as shown in Figure 3. This is documented also in Fig. 4, showing SEM image.

This requires significant reviewing of the manuscript along this line, in particular title change.

Other minor changes:

Line 60: representative example; the monolayer Aurivillius oxide Bi2WO6 has a  sandwich substructure of [BiO]+ – [WO4]2- – [BiO]+ mimicking heterojunction interface  with space charge (???). What authors mean?

Line 66: photocatalytic degradation of organic dye. What dye

Line 77: Two-dimensional (2D) Bi2WO6  monolayers showed antibacterial efficiency in both studied strains (gram-negative and gram-positive bacteria) without the presence of light (in the dark), showing a dimension-dependent antibacterial activity of Bi2WO6 system and improving its properties in 80 relation to the bulk material. What is the “dimension-dependendency”???

Figure 5: (a) Raman and (b) FTIR spectra of Bi2WO6 monolayers at room temperature (see above)

Line 184: Modulation of antibiotic activity by Bi2WO6 monolayers  (see above)

Etc.

Author Response

We thank the referee for the careful reading of the manuscript and his/her valuable comments. We follow his/her suggestions as far as possible, and we improved the manuscript accordingly. In the following, we give a detailed response to all individual comments.

Authors: We thank the referee for his/her comments/recommendations/suggestions on our manuscript.

First question is what sort of material have authors manufactured. Moreover, manuscript uses not very precise language that is hard to understand. As seen in AFM image (Fig. 2) the produced material is not “monolayer” but “multilayered” material oriented during deposition along the same crystallographic direction. On contrary to the original paper by Yangen Zhou from Nature Comm. 2015, authors have not provided any experimental evidence that it is in fact monolayer. Moreover, to reference to the someone else paper is not enough and authors should provide in the manuscript exact experimental procedure and experimental data to convince us what kind of material are they working with. If this is not possible, since the deposited material has quite limited thickness and contains only several sandwich-type layers, probably the best description would be “nanolayered material”. Hydrothermal methods in general, allow to obtain homogenous monocrystalline multilayered solids of almost unified size and the structure as shown in Figure 3. This is documented also in Fig. 4, showing SEM image.

Response: We thank the referee comment. We replaced the description "Bi2WO6 monolayer" with "layered Bi2WO6nanosheets" throughout the text, including the title. In addition, we inserted the synthesis procedure in the revised version of the manuscript.

  • Line 60: representative example; the monolayer Aurivillius oxide Bi2WO6 has a sandwich substructure of [BiO]+ – [WO4]2- – [BiO]+ mimicking heterojunction interface  with space charge (???). What authors mean?

Response: We thank the referee comment. This information has been corrected.

  • Line 66: photocatalytic degradation of organic dye. What dye.

Response: We thank the referee comment. This information was included in the article.

  • Line 77: Two-dimensional (2D) Bi2WO6  monolayers showed antibacterial efficiency in both studied strains (gram-negative and gram-positive bacteria) without the presence of light (in the dark), showing a dimension-dependent antibacterial activity of Bi2WO6 system and improving its properties in 80 relation to the bulk material. What is the “dimension-dependendency”???

Response: We thank the referee comment. Bi2WO6 with three-dimensional morphology (Bulk or nanoparticle) does not present antibacterial action without visible light irradiation. On the other hand, the synthesis of the two-dimensional material promoted Bi2WO6 an antibacterial performance without the need for visible light irradiation. The antibacterial activity without the presence of visible light is related to the variation of the material's dimension (change from 3D to 2D).

Reviewer 2 Report

Pretty good work. I personally am more interested in the progress you mentioned, from line 174 to 177, it could be better if a flow chart is presented.

Author Response

We thank the referee for the careful reading of the manuscript and his/her valuable comments. We follow his/her suggestions as far as possible, and we improved the manuscript accordingly. In the following, we give a detailed response to all individual comments.

Pretty good work. I personally am more interested in the progress you mentioned, from line 174 to 177, it could be better if a flow chart is presented.

Authors: We thank the referee for his/her comments/recommendations/suggestions on our manuscript.

Reviewer 3 Report

In this manuscript, authors reported using a existing photocatalytic monolayer Bi2WO6 material for antibacterial treatment. Material characterization partially verifies the synthesis of Bi2WO6 material yet the monolayer structure needs further verification. In vitro experiments show good inhibition efficacy on both Gram-negative and -positive bacteria. It is suggested to do more experiments to solidify this work.

1. The thickness of layered Bi2WO6 looks inconsistent from SEM (Figure 4) and AFM (Figure 2): thicker in SEM images. Also the height profile of AFM in figure 2 looks odd. Please provide AFM image as well as corresponding height profile with better quality and also analyze the thickness in SEM and generate a histogram of statistical analysis.
2. Some experimental design has to be optimized. For example, In the antibiotic experiments as shown in Figure 7, authors have to provide Bi2WO6 side-by-side comparison as dark control to demonstrate that the antibiotic efficacy came from material-mediated photocatalysis exclusively.
3. Some data qualities need to be improved. For example, multiple columns in Figure 6 and 7 seem no error bar. Please give reasonable explanation. 
4. Authors need to carefully polish the language and amend typose. For example, Bi2WO6 in legends of figure 7 A-C as well as Line 225. Table 1 looks messed up.

Author Response

We thank the referee for the careful reading of the manuscript and his/her valuable comments. We follow his/her suggestions as far as possible, and we improved the manuscript accordingly. In the following, we give a detailed response to all individual comments.

In this manuscript, authors reported using a existing photocatalytic monolayer Bi2WO6 material for antibacterial treatment. Material characterization partially verifies the synthesis of Bi2WO6 material yet the monolayer structure needs further verification. In vitro experiments show good inhibition efficacy on both Gram-negative and -positive bacteria. It is suggested to do more experiments to solidify this work.

Authors: We thank the referee for his/her comments/recommendations/suggestions on our manuscript.

The thickness of layered Bi2WO6 looks inconsistent from SEM (Figure 4) and AFM (Figure 2): thicker in SEM images. Also the height profile of AFM in figure 2 looks odd. Please provide AFM image as well as corresponding height profile with better quality and also analyze the thickness in SEM and generate a histogram of statistical analysis.

Response: We thank the referee comment. To perform SEM images, the sample was dispersed by ultrasound in ethanol and dropped onto a glass substrate. Due to the very low thickness of the nanolayers, they naturally tend to have a crumpled paper-type morphology. Furthermore, measuring the thickness of nanolayers by SEM images is a very difficult task using a microscope with a beam of electrons emitted by a tungsten filament. Metallization mechanisms to obtain better SEM image quality becomes unfeasible since this procedure would modify the thickness of crumpled nanolayers. However, in the SEM image shown in the manuscript we see that the nanosheets have uniform morphology and very low thickness, but the image quality makes it impossible to measure the thickness of the layered Bi2WO6 nanosheets. The roughness of the glass substrate and the very low thickness of the nanolayers make it difficult to perform the AFM image, but even with all the difficulties, the AFM analysis showed the very low thickness of the material, and confirmed the success of the synthesis process.

Some experimental design has to be optimized. For example, In the antibiotic experiments as shown in Figure 7, authors have to provide Bi2WO6 side-by-side comparison as dark control to demonstrate that the antibiotic efficacy came from material-mediated photocatalysis exclusively.

Response: We thank the referee comment. Figure 6 shows the data regarding the modulation tests in the dark, where the material interacted antagonistically with the antibiotics. Figure 7 shows the results of modulation with light exposure, where the antibiotics were not effective, i.e., the material did not improve the effect of these drugs, because due to the photocatalytic effect, there was probably a degradation of the drugs, which justifies the antagonistic results

Some data qualities need to be improved. For example, multiple columns in Figure 6 and 7 seem no error bar. Please give reasonable explanation.

Response: The referee is correct, this information was inserted in Figure 6 and 7 as was requested.

Authors need to carefully polish the language and amend typose. For example, Bi2WO6 in legends of figure 7 A-C as well as Line 225. Table 1 looks messed up.

Response: We thank the referee comment. A review of writing and language was performed throughout the manuscript.

Round 2

Reviewer 1 Report

Manuscript can be accepted for publication.

Author Response

We thank the referee for the careful reading of the manuscript and his/her valuable comments. 

Reviewer 3 Report

I am still questioning the height profile of AFM. There is no additional experiments or additional data analysis done for the revision.

Author Response

We thank the referee comment. We've added a new Figure 2 containing AFM results. We inserted new height profiles measured at different points of the layered Bi2WO6 nanosheets.

Round 3

Reviewer 3 Report

The reviewer agree to endorse the manuscript for publication.